# Acute Hamstring Injury Prevention Programs in Eleven-a-Side Football Players Based on Physical Exercises: Systematic Review

**DOI:** 10.3390/jcm10092029

**Published:** 2021-05-09

**Authors:** Adolfo Rosado-Portillo, Gema Chamorro-Moriana, Gloria Gonzalez-Medina, Veronica Perez-Cabezas

**Affiliations:** 1Department of Physiotherapy, University of Seville, 41009 Seville, Spain; adolforosadoportillo@gmail.com; 2Head of Research Group “Area of Physiotherapy” CTS-305, Department of Physiotherapy, University of Seville, 41009 Seville, Spain; 3Research Group “Physiotherapy and Health” CTS 986, Department of Nursing and Physiotherapy, University of Cadiz, 11009 Cadiz, Spain; gloriagonzalez.medina@uca.es; 4Research Group “Empowering Health by Physical Activity, Exercise and Nutrition” [CTS1038], Department of Nursing and Physiotherapy, University of Cadiz, 11009 Cadiz, Spain; veronica.perezcabezas@uca.es

**Keywords:** physical exercises, physical therapy, health, eccentric contraction, strength training, muscular diseases, prevention, hamstring muscles, football, soccer

## Abstract

(1) Objective: To analyze the exercise programs used to prevent of acute hamstring injuries in eleven-a-side football players, and their effectiveness. (2) Methods: A systematic review (PRISMA) was conducted (2008–2020), including RCTs, that exclusively used physical exercises as a prevention method. (3) Results: Ten studies were selected considering 14 interventions, including nine different programs: FIFA11+ (11+), Harmoknee, eccentric Nordic Hamstring Exercise (NHE) exclusively, with eccentric exercises, with stretching or with proprioceptive, New Warm-up Program (NWP), Bounding Exercise Program (BEP), the only one with no positive results, and proprioceptive exercises. Incidence of injuries and strength were the most considered variables, both with favorable evidences. Programs including NHE, which assessed injury incidence, were always effective. The 11+ program was effective in injury incidence and strength; NWP was effective in balance, stability, and strength. (4) Conclusions: The exercise programs discussed were effective to prevent acute hamstring injuries in football players except BEP and partially Harmoknee. Exercises mostly used to reduce the risk of hamstring injuries are those of eccentric force due to its functionality, especially NHE. Only concentric contractions and isometric contractions obtained significant favorable results. The most complete and promising programs were 11+ (in injury incidence and strength) and NWP (strength, balance, and stability). NWP was the best in strength.

## 1. Introduction

Lower limb injuries stand out in sports such as football, where they account for 92% of the total [1,2,3]. The physical demands and characteristics of this sport lead to a high incidence of injuries [4]. For this reason, we have felt it necessary to recall some of the many and interrelated intrinsic risk factors. Thus, nutrition [5], sleep deficit [6], gender [7], physical condition [8], previous injuries [9], etc., among others, are determining factors in injuries, especially in lower limbs.

In football, it is worth noting the contingency factor or the control of contingencies that occur continuously in competition and that influence the incidence of injuries [4] such as contusions from contact with opponents or falls, joint injuries to the ankles and knee due to twists or poor support on the ground and musculoskeletal injuries (quadriceps, hamstrings, calves, adductors) from constant starts, stops, sprints, ball kicks, jumps, etc.

There are additional influential aspects such as dehydration during training or competitions, poorly planned warm-ups or stretching, progressions in exercise intensity, rest times throughout the season (training overload), level of physical qualities [4] (endurance, strength, speed, motion range, and coordination), etc. Therefore, the technical team plays a fundamental role in controlling and preventing injuries [10].

Focusing on hamstring muscle injuries and strength quality, we also need to consider the type of muscular contraction developed (concentric, eccentric, and isometric) and the type of force, especially explosive force, for example starting [11]. Thus, the position the player has, involving frequent characteristic actions, will be another risk factor. Although the force level factor seems obvious, the imbalance of forces between agonist–antagonist could influence even more [12], i.e., hamstrings–quadriceps.

Another essential factor is muscle fatigue [11], which is related to the training plan or poor nutrition, among others. Due to the specific technical gestures of this sport activity and the muscular fatigue it causes, there are structures that tend to be injured more frequently; for example, 37% of the injuries are related to the hamstring muscle mass [2], while in 53%, it is the biceps femoris muscle that is affected [8]. Among the hamstring pathologies, acute injuries are more relevant due to their regularity, among which the following stand out: partial ruptures with an incidence of 94%, contusions, tendonitis, full ruptures, etc. [8]. In fact, muscle strains represent 30% of the injuries, and 28% of these strains involved the hamstrings [13]. In addition, these acute injuries can be complicated by the degree of recurrence they have, between 12 and 33% [14].

Numerous studies [12,15,16,17] advocate preventive programs based on physical exercise, as they are considered the most effective methods [17]. This is due, amongst other reasons, to the fact that movement is an essential component for well-being, producing physical changes that condition health [18].

Within this physical therapy, training focused on eccentric force is emphasized, which implies movement due to the frequency with which it is used and the benefits it provides [9,19]. Strength together with speed are the main variables that increase the risk of suffering an acute hamstring injury [18]. Authors such as Petersen et al. [9] affirm in fact that the higher the speed, the greater the strength required. Guex et al. [13] add that football players who have a high risk for injuries are those who play in speed position.

Finally, we should not forget that the motion range quality (e.g., hip flexion and knee) is a risk factor that can lead to hamstring shortening, and in turn, to fibrillar injuries. Such exposed risk factors will be addressed throughout this document, not only by the authors of the selected studies in order to justify their prevention interventions and variables analyzed, but also by the authors of this document, who will discuss them below.

Therefore, the consideration of the risk factors and the prevention of this characteristic health problem in footballs plays are fundamental [14], not only because of its incidence but also because of the high economic cost to the clubs [20] and the long treatment involved in their functional recovery [14]. All this consequently causes a great loss in sports practice [21].

Thus, the present systematic review aims to analyze the exercise programs used for the prevention of acute hamstring injuries in football players, as well as their effectiveness.

## 2. Materials and Methods

A systematic review was carried out based on the PRISMA protocol [22]. The checklist is provided in Appendix A.

### 2.1. Data Sources and Search Strategy

The systematic review included an electronic search of Publisher Medline (PUBMED), Cumulative Index to Nursing and Allied Health Literature (CINHAL), SCOPUS, and SPORT DISCUS. The search was carried out for the last 12 years, including 2020. The following medical subject headings (MeSH) terms were used: “injury”, “hamstring”, “soccer” (football), and “prevention”. Other terms of interest were also included: “damage”, due to its frequency in scientific quality studies; and soccer to differentiate football 11 from Australian, American, etc. Those have been discussed scientifically in numerous studies.

The search strategy used was: (injury OR damage) AND hamstring AND (soccer OR football) AND prevent *.

### 2.2. Selection of Studies and Inclusion Criteria

The studies included in this review needed to meet the following criteria:Studies that use only exercise programs for the prevention of acute injuries in football players, that is, without complementing other preventive methods.Studies with samples of healthy football players.Study design: Controlled and Randomized Controlled Trials (RTC).Language: English.

The titles and abstracts of the search results were screened to check if a study met the pre-established inclusion criteria. We obtained the full text article of the studies that met the criteria and documented the causes for any exclusions at this stage.

### 2.3. Data Extraction

Data extraction was carried out by one reviewer (A.R.) and checked for accuracy by a second reviewer (V.P.) using a table designed to detail information on study features, participant characteristics, interventions, variables, assessment tools, and outcome measurements. Disagreements between reviewers were resolved by consensus. If consensus was not reached, the final decision was made by a third reviewer (G.C.). The reviewers were not blinded to authors, date of publication, and journal publication. 

### 2.4. Quality Appraisal

Apposite studies were assessed for methodological quality using the Physiotherapy Evidence Database (PEDro) critical appraisal tool [23]. This method was valid and reliable for assessing the internal validity of a study (criteria 2–9). We also evaluated the adequacy of statistical information for interpreting the results (criteria 10–11) [24,25,26]. PEDro consists of 11 criteria overall; although criterion 1 refers to the external validity of trial and is not included in the final score [26]. Each criterion could be Yes (one point) or No (0 points), with a maximum score of ten. Only “fair” (scores 4/5) and “high” (scores 6/10) quality studies were included in this review.

## 3. Results

We found 923 articles in the electronic database. Following the removal of duplicates, 298 articles were screened by title, abstract, and full-text, according to not being RTCs and not being healthy football players. After screening, 10 articles were included in this review. Figure 1 shows the search and study selection process.

### 3.1. Characteristics of Included Studies

The features and results of the included papers in this review are shown in Table 1. 

### 3.2. Quality Assessment

The PEDro Scale has been used as a valid measure of the methodological quality of clinical trials [24]. All selected papers rated as “fair” and “high” quality (>4). The results of the PEDro scale are shown in Table 2. 

The items “Measures of at least one key outcome were obtained from more than 85% of the subjects initially allocated to groups” (9); “All subjects for whom outcome measures were available received the treatment or control condition as allocated or, where this was not the case, data for at least one key outcome was analyzed by ‘intention-to-treat’” (10); and “The results of between-group statistical comparisons are reported for at least one key outcome” (11) were scored by all papers. Although the studies were considered to be of “fair” and “high” quality, there were two items with 0 scores: “Blinding of all subjects” (5) and “Blinding of all therapists who administered the therapy” (6) for all ten papers. Furthermore, the items “Groups were similar at baseline regarding the most important prognostic indicators” (4), and “Measures of at least one key outcome were obtained from more than 85% of the subjects initially allocated to groups” (8) only two papers did not score.

### 3.3. Participant Characteristics

In all the included studies, the participants were healthy males, although they may have suffered acute hamstring injuries in the past, with ages ranging between 17 and 40 years, they were professional or amateur football players representing American and European leagues. All articles were published between 2010 and 2020. Five of the 10 articles [12,15,29,30,31] specified that they could not have suffered a previous injury in lower limbs [12,15] or only in the last 6 months [29,31] (Nacleiro et al., 2013; as exclusion criteria). Three of them [16,29,31] excluded players who participated in lower limb injury prevention programs in previous seasons. In five of the 10 studies [9,16,19,27,30], the sample chosen was over one hundred. Characteristics of the 10 included studies are summarized in Table 1.

### 3.4. Intervention

All the selected studies used a period of application of their intervention methods of between 6 weeks [29] and one full season [16].

Based on the inclusion criteria of this review, the intervention programs used by all the authors consisted of physical exercises. Van de Hoef et al. [27] used two experimental groups that were continuing regular football training. In addition, the intervention group was performing a 12-week Bounding Exercise Program (BEP). This consists of a gradual build up of a maintenance program for the entire football season. The primary outcome was hamstring injury incidence. The secondary outcome was compliance with the BEP during the football season and 3 months thereafter. This program consists of a gradual build up to plyometric exercises; it is a specific injury prevention program. It can be easily implemented during warm-up and is expected to improve sprint and jumping performance [27]. Two of the studies [9,19] exclusively used the Nordic exercise (NHE), increasing the repetitions and series progressive every week: one of them completed it in 25 sessions [19] and the other completed it in 27 [9]. The NHE is an eccentric exercise that consists of placing oneself in a kneeling position, fixing the ankles with the help of a partner and performing a slow and controlled lowering of the trunk until touching the ground with the chest [9]. The research by Sebelien et al. [30] applied the NHE together with hamstring stretches, performing both in all workouts. Naclerio et al. [31] prescribed three exercises to IG: two proprioceptive and the NHE. While this same author [29] used two experimental groups, one only performed eccentric exercises (including the NHE) and the other performed proprioceptive exercises. The intervention period of these last two programs was 18 sessions. Finally, the remaining four studies [12,15,16,27] used a program, called 11+, previously Federation International de Football Association 11+ Method, (FIFA 11+). This method is an injury prevention program designed as alternative warm-up program to address lower extremity injury in football. It is a 20-min program that is used on the field without any additional equipment. It consists of 15 exercises divided into three groups: running exercises (8 min) that encompass cutting, decelerating, change of direction, and proper landing techniques; strength, plyometric, and balance exercises (10 min); and specific running exercises (2 min) to prepare the player for athletic participation [12,15,16]. Silvers-Granelli et al. [16] only used this method, while Daneshjoo et al. [12] and Daneshjoo et al. [15] also assessed the Harmoknee intervention program. This training protocol (20 to 25 min) consists of five parts: warm up, muscle activation, balance, strength (NHE), and core [12,15]. Both the 11+ and the Harmoknee programs also incorporate the eccentric NHE. Therefore, six of the seven programs addressed in this review apply NHE, although all studies considered eccentric exercises. In addition Ghareeb et al. [28] assessed the New Warm-up Program (NWP) for balance and isokinetic strength of the quadriceps and hamstrings at 60, 180, and 300 degrees per second. The participants practiced for 6 weeks during one football season, one group with NWP and the other one with 11+.

Figure 2 represents the frequency with which these programs appear in the selected studies.

### 3.5. Assessment Tools

The tools used for the evaluations, in order of highest to lowest frequency, were weekly questionnaires for injury incidence and time lost due to injury incidence [9,10,16,27,30], three of them were ad hoc [10,27,30]; isokinetic systems, Biodex 3 Isokinetic Dynamometer [12,15,28] and Cybex 6000^®^ [30]; load cell and I METRIC V. 8.32 software^®^ for isometric contraction [29,31]; Biodex Balance System SD [28]; Athlete Single Leg Stability Test protocol [28]; and a 40-meter sprint test [30].

### 3.6. Results of Each Article

Four of the 10 articles [9,16,19,30] obtained a significant difference in relation to the number of injuries, thus demonstrating the effectiveness of the 11+ prevention program [16], NHE exercises isolated [9,19], and NHE + stretches exercises [30]. On the other hand, van de Hoef et al. [27] found no evidence that a new functional injury prevention exercise program (BEP) prevented hamstring injuries in adult male amateur football players. Sebelien et al. [30] showed a significant decrease between both groups in the injury incidence. However, the measures of hamstrings eccentric strength, sprint speed, and quadriceps concentric strength did not obtain significant differences. In their two articles, Naclerio et al. [29,31] found that after the execution of eccentric strength exercises, isometric force was increased in the smallest angles of knee flexion, 45° being significantly greater. On the contrary, proprioceptive exercises [29] increased isometric strength at 80°. When eccentric strength exercises were combined with proprioception exercises [31], the significant increase was also at 80°. Ghareeb et al. [28] used the NWP program to address the handicaps of the 11+ warm-up program, particularly to improve balance and strength. There is evidence that supports that a neuromuscular training program of jumping, balance, agility training, and landing mechanics can decrease lower extremity injury rates among athletes Although we did not measure injury rates among the group, the NWP incorporates these training elements and can possibly be beneficial in reducing injuries [10]. Their results were not significantly different regarding balance and stability. Finally, the remaining [12,15] obtained a significant increase in the hamstring concentric strength after executing 11+. Therefore, this program was effective in avoiding acute hamstring injuries. In contrast, the Harmoknee program [12,15] showed significant increases in hamstring and quadriceps strength in one of the articles [15] but not in the other. The significance and effectiveness of prevention protocols are shown in Table 3.

## 4. Discussion

This systematic review addresses the randomized clinical trials of the last 12 years that applied exercise programs for the prevention of acute hamstring injuries in football players. Thus, it analyzes the characteristics of the participants and the intervention methods employed, the outcome measures applied, the variables considered, and the results of the selected studies applied, among others.

As for the samples analyzed in the review, all are from men, although the injuries they can suffer do not have the same prevalence as for women [7]. Some relevant anthropometric and physiological differences between the sexes, not only fitness levels (highly influenced by training) should be considered, such as height, body (forearm), foot length/shoe size, speed, endurance, football endurance, % muscle, lower leg strength, jumping height, and kicking velocity [8]. Therefore, the most commonly used prevention methods in women’s football may be different to that of men’s. On the other hand, the samples corresponded to professional and amateur football players, while the physical requirement between both leagues vary significantly [32]. There is evidence that additional eccentric hamstring exercise decreased the rate of overall, new, and recurrent acute hamstring injuries, in male football players, both professional and amateur ones [9,33].

The physical exercises approached depend on a multidisciplinary team that plays a key role in the prevention of injuries in football [19]. This is formed by the Athlete, Team Coach, Physical Therapist, Sports Physician, and Fitness Trainer [10]. They intervene directly or indirectly in the prescription, application, and/or supervision of the exercises of the programs so that their execution is correct and therefore effective. The clinical criteria of the multidisciplinary team allow the injured user to return to playing (RTP) or training [10]. The figure of the physiotherapist stands out in these functions, especially in the supervision of these exercises, according to various authors [9,16,19]. The prevention work of the physiotherapist is even recognized by the World Confederation of Physical Therapy, which defines physiotherapy as the set of techniques that through the application of physical agents, cure, prevent, recover, and readapt patients susceptible to receiving physical treatment, that is, therapeutic physical exercise [18].

In relation to the physical exercises used in the selected studies, those for strengthening the hamstring muscle using eccentric contractions stand out. These are employed in all studies. Among these strengthening exercises, the Nordic Hamstring Exercise (NHE) is present in 66.7% of the prevention programs used. In spite of that, this exercise was only exclusively carried out in of 22.2% [9,19], and the other 44.4% of programs combined the NHE with other exercises [12,15,16,29,30,31]. Only one study [29] used proprioceptive exercises alone in one of its interventions. According to the results of this review, the exclusive NHE and 11+ were targeted as the most used prevention programs for acute hamstring injuries in football players, which is possibly because of their effectiveness. Due to the heterogeneity of the samples that use these interventions, this document cannot highlight the effectiveness of one over another. Although both programs are effective, it should be noted that the NHE is a single exercise, while the 11+ consists of a complete program that includes, in addition to the NHE, a general warm-up, specific strength exercises for the core, and a subsequent warm-up that is more related to the sport in question [16]. This program was executed by 40.0% of studies [12,15,16,28].

Continuing with effectiveness, it is difficult to comparatively analyze and associate the characteristics (content, dosage) of the nine prevention programs addressed in this review with their benefits, due to the heterogeneity mentioned above. However, in the following, we will cautiously make some interesting observations.

All the programs that used the NHE exclusively or complemented it with other exercises [9,12,15,16,19,28,29,30,31] obtained positive results in one of the variables analyzed. Similarly, all the programs that used NHE and assessed the incidence of hamstring injuries [9,16,19,30] were effective.

Among the programs that did not contain NHE, BEP [27] stood out negatively. It was the only one that did not obtain any positive results despite being based on plyometric exercises (jumping), which are highly valued for their efficacy in the sports and physiotherapeutic field [34]. Perhaps the exclusivity of these exercises is the great difference regarding other programs that contemplate them.

All nine programs covered include strength exercises. Even the proprioception only exercise program indirectly works on isometric strength, achieving a partial improvement on hamstrings [29], but only five of the nine programs assessed strength: 11+ [12,15,28], Harmoknee [12,15], NHE + eccentric [29], NHE + propioc [31], NHE + stretches [30]. Of the five programs that tested it, NHE + stretches [30] and Harmoknee in one of the two cases [12] failed to increase it. Proprioception (mainly balance) is the second most used type of exercise (11+ [12,15,16,28], NWP [28], Harmoknee [12,15], NHE + propioceptive [31], proprioceptives [29] and NHE [9,19]). Stretches are included in four of them: NWP [28], Harmoknee [12,15], 11+ [12,15,16,28], and NHE + stretches [30] as well as velocity (11+ [12,15,16,28], BEP [27], Harmoknee [12,15], and NWP [28]) and range of motion (NWP [28], Harmoknee [12,15], 11+ [12,15,16,28], proprioception + stretches). Finally, three programs include endurance (Harmoknee [12,15], 11+ [12,15,16,28], and NWP [28]).

It follows that the most comprehensive programs are 11+, Harmoknee, and NWP, all of which share the addressing of all physical qualities, i.e., intrinsic risk factors [4] for hamstring injuries (see introduction). Of these programs, the efficacy of the 11+ in relation to the variables hamstring injury incidence [16] and strength [12,15,28] is noteworthy, although it is not significant in balance and stability [28]. However, NWP assesses and is effective for balance, stability, and strength [28], although NHE is not used [28]. These data are not decisive as they are only offered by one study [28]. It is striking that Harmoknee only assesses the strength variable despite working on a large number of different types of exercise [12,15]. Furthermore, improvements in strength are found in only one of the two cases [15].

Based on the results of this review, the most promising level of effectiveness corresponds to 11+ [12,15,16,28] and NWP [28]. Both are similar and comprehensive warm-up programs, given the variety of exercises they include. However, some differences between them can be highlighted. Both take into account the risk factor related to exercise intensity progressions. NWP contemplates six levels, while 11+ contemplates only three. Although NWP seems more advantageous in principle, 11+ takes into account the basic principle of applying intensity on an individual basis, i.e., each player progresses in intensity according to their qualities [35]. In contrast, NWP [28] progresses in intensity for the group as a whole. On the other hand, NWP [28] does not include the star exercise of this review, the NHE. However, it integrates specific exercises to improve the deficiencies detected in the 11+ regarding balance, using destabilizing materials such as Bosu^®^, disc pillow, and mini-trampoline; and strength, incorporating stairs and obstacles [28]. Thus, NWP is effective in improving balance in contrast to 11+ [28]. In terms of strength, NWP achieved a more significant improvement than 11+ [28].

When addressing efficacy in the prevention of acute hamstring injuries, we can differentiate the prevention of new onset lesions and recurrent injuries. Thus, five of the studies [9,16,19,27,30] measured the prevalence of acute hamstring injuries, but only that of Petersen et al. [9] compared the effectiveness of a prevention program between new onset injuries and recurrent injuries. They themselves [9] showed that the use of NHE exclusively decreased the probability of suffering an acute recurrent hamstring injury by 85%, compared to 60% of new onset injuries. 

In contrast to the selected studies, Grooms et al. [36] affirm that the risk of lower limb injuries in football could be reduced by a structured warming program based on balance exercises, neuromuscular control, and muscle strength.

In agreement with the previous paragraph, Mendiguchia et al. [37] gambled, in their latest study, on a treatment protocol after a partial break in the hamstrings where NHE was present, to reduce the risk of a recurrent injury. According to this author, the eccentric strength of hamstrings improves, and consequently, so does its functionality [37] through this exercise. Therefore, this exercise can be used not only in prevention protocols but also in the treatment of injuries.

Lepley and Butterfield [38], similar to Mendiguchia et al. [37] and the authors of this review, advocate the effectiveness of eccentric exercises, since this kind of contraction entails more advantages than others (concentric and isometric). In fact, it is the most functional contraction, it promotes the growth of sarcomeres in series at an optimized metabolic cost, it increases neural excitability, and consequently, it improves intramuscular coordination [38]. Moreover, hamstring injury mechanisms in football, which usually involve eccentric performances, imply high tension in the muscle fibers. Thus, the hamstrings should get used to eccentric contractions in training to prevent injuries.

On the other hand, in contrast to the ideas put forward by Van Horst et al. [10] and Petersen et al. [9], the study by Sebelien et al. [30] assured that the eccentric strength of the hamstring muscle decreased by performing the NHE despite the reduction in the risk of injury recorded. This seems contradictory, since considering the injury mechanism, if we increase the eccentric strength of the hamstring muscle, the probability of injury will decrease. It needs to be borne in mind that the degree of reliability of this study [30] was very low, because the complete measures collected were a minimum part of the total sample (22.7%).

Among the variables analyzed by the selected studies, the following stand out: the incidence of the injury [9,16,19,27,30] collected and measured in a similar way by each study, as well as strength. The latter has been evaluated based on the type of contraction: concentric strength in quadriceps and hamstrings [12,15,28,30], isometric strength in quadriceps [30] and hamstrings [29,30,31], and eccentric force in quadriceps and hamstrings [30]. It should be noted that concentric contractions were the most evaluated even though eccentric contractions are the key to most interventions as mentioned above. Sebelien et al. [25] were the only ones who assessed eccentric contraction with no positive outcomes. On the basis of the injury mechanism, the use of the variable “eccentric force” would be more convenient [8]. In contrast, according to Van Hooren et al. [39], the hamstring mass undergoes an isometric contraction while running in the open kinetic chain change (last phase of the oscillation) to the closed kinetic chain (first support phase), exactly at the moment of most susceptible change to injury. Therefore, this author highlights the importance of influencing isometric strength for the prevention of acute hamstring injuries [39]. 

Naclerio et al. [31] and Naclerio et al. [29] demonstrated the efficacy of their intervention program in isometric hamstring strength. Daneshjoo et al. [12], Daneshjoo et al. [15], and Ghareeb et al. [22] demonstrated it for the concentric strength in quadriceps and hamstrings. 

On the other hand, only Daneshjoo et al. [12], Daneshjoo et al., Ghereeb et al. [28], and Sebelien et al. [30] considered the equivalence of hamstring–quadriceps force. This muscular balance between agonist and antagonist is presented as one of the determining risk factors not only in acute hamstring injuries but also in knee injuries in general [12]. No other study takes this risk factor into account, since the authors who addressed the force variable focused their attention on the isometric force of only the hamstring mass [29,31].

Considering the resulting measures, it was observed that technology-based assessment tools were the most used. Isokinetic [12,15,28,30] were the best option for the precise calculation of the most considered variable, i.e., strength, in its three aspects: eccentric, concentric, and isometric. Additionally, load cells were employed to assess isometric strength [29,31]. Technology was also used to assess stability and balance [28]. However, the applications (ad hoc or standard) were used on numerous occasions to collect the number of acute hamstring injuries, their severity and the time lost due to injuries [9,10,16,27,30]. According to Chamorro-Moriana et al. [40], the authors of this study recognize that technological progress has led to the development of highly useful tools in the field of functional recoveries and preventions that complement conventional tests. Despite the multiple benefits that new technologies offer, a physiotherapist’s face-to-face test of a patient cannot be equaled by technological means. Observation and the personalized and intuitive adaptation of the health-care professional are key to a successful prevention program [40].

Regarding other aspects that could influence acute hamstring injuries such as nutrition, hydration [41], fatigue [21], etc., they were not addressed in any of the selected papers. Therefore, we understand that the results obtained by the studies may have certain limitations in this regard. It is true that considering a large number of risk factors would exponentially increase the difficulty of the study.

The authors decided to carry out a meta-analysis of the studies that used the same exercise programs, either because of their proven efficacy or because of their frequency. However, this review considered some handicaps such as the lack of pre-test, post-test applied at different times, varied doses, and duration of the program. In addition, the heterogeneity of the variables analyzed and the assessment of the different types of strength in hamstrings implied a limitation to make more conclusive comparisons of the results obtained. Therefore, the research did not include a meta-analysis complementary to the review. Prospective to the results and limitations found in the selected studies, we believe that RCTs should be carried out with similar objectives but considering other influential factors such as those mentioned in the discussion.

## 5. Conclusions

All the exercise programs discussed in this review, with the exception of BEP and the Harmoknee in one of the studies, were always effective as preventive methods of acute hamstring injuries in football players.

Among the exercises applied to the hamstring, those based on the eccentric strength of the hamstring stood out significantly due to their functionality. Of those, the most used was NHE, whether applied exclusively or with other exercises. However, eccentric contractions were assessed in only one case and with no favorable results. Concentric contractions, the most frequently assessed, and isometric contractions obtained significant favorable results. Only two authors considered the muscle balance between agonist and antagonist of interest. The 11+ prevention program, which includes the NHE, was the most widely applied method, followed by the Harmoknee (which also include the NHE) and the exclusively used NHE.

The programs using the NHE obtained positive results in some of the variables analyzed. Incidence of injury and strength were the most considered variables. Programs using NHE and assessing the incidence of hamstring injuries were always effective. The NHE alone reduced recurrent injuries versus original injuries to a greater extent. Of the programs including NHE, 11+, Harmoknee, NHE + eccentric, and NHE + proprioception obtained strength improvements.

The most comprehensive programs in terms of exercise variety and promise were 11+ and NWP. Of these, the efficacy of 11+ regarding the incidence of hamstring injuries and strength is noteworthy, although it is not significant for balance and stability. NWP is effective for balance, stability, and strength. NWP achieved a more significant improvement regarding strength than 11+.

Technology-based assessment systems, especially isokinetic systems, stand out for their usefulness and efficiency in this type of trial.

This review provides the football coaching staff with an effective tool for the prevention of acute hamstring injuries. The evidence suggests that eccentric contractions appear to be the key to interventions, and it is recommended that they are not only developed but also assessed. 

## Figures and Tables

**Figure 1 jcm-10-02029-f001:**
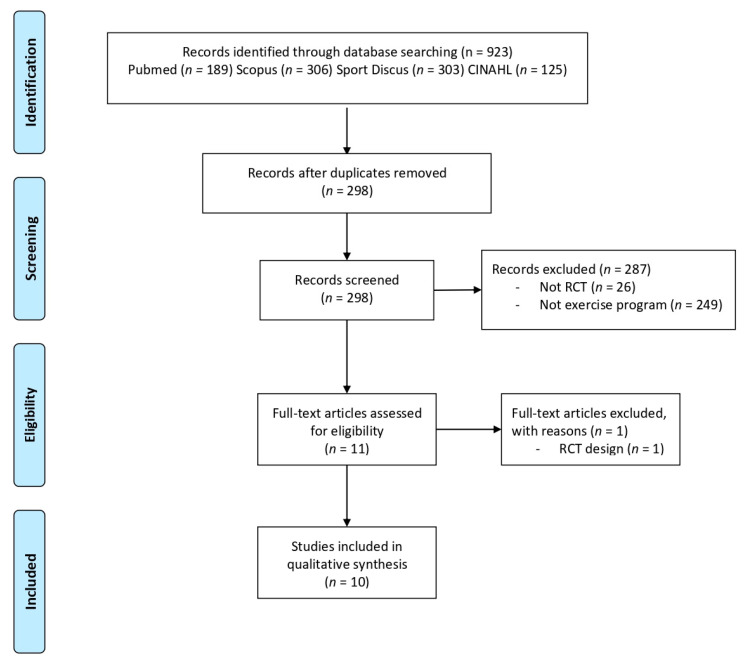
PRISMA flow diagram.

**Figure 2 jcm-10-02029-f002:**
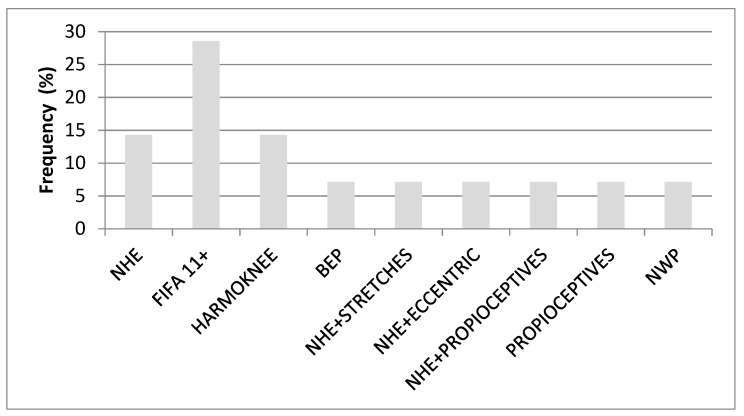
Frequency with which the intervention programs were employed in the selected studies. Abbreviations: NHE: Nordic Hamstring Exercise. FIFA: Federation International de Football Association. BEP: Bounding Exercise Program. NWP: New Warm-up Program.

**Table 1 jcm-10-02029-t001:** Characteristics of the selected studies.

References/ Objectives	Participant Characteristics n/Age	Interventions	Assessment Tools	Outcomes
	Inclusion/Exclusion Criteria		Variables	
Van de Hoef et al. [27]. (2019). To examine the preventive effect of the bounding exercise program on hamstring injuries in amateur football players.	*n* = 400 (≈16 players × 32 teams) (GC:171; IG:229) Year 2016–2017 Male, health Age = 18–45 years	CG: practice as usual. IG: BEP program for 39 weeks: walking lunges, triplings followed by drop lunges, and bounding.	Ad hoc registration forms.	95 % IC Injury incidence CG: 1.39/1000 h IG: 1.12/1000 h No statistically significant (OR = 0.89, 0.46–1.75) Severity between the groups (*p* > 0.48). Compliance with BEP: 71%
	IC: Football teams from Dutch first class amateur competition EC: joined a team after the start of the trial.		Hamstring injuries incidence/1000 football hours, severity injuries and compliance (BEP)	
Ghareeb et al. [28]. (2017). To examine the effectiveness of the NWP as compared to the FIFA11+ program in high school-aged football players as they effect the development of balance and strength.	*n* = 34 Age IG1(17) = 16.53 ± 1.125 years, IG2(17) = 16.53 ± 1.068 years	IG1: NWP program. IG2: 11+, 20-min, 3 sessions/week, 6 weeks.	Biodex Balance System SD (stood on the each leg). Athlete Single Leg Stability Test protocol (anterior/posterior and medial/lateral). Biodex Isokinetic Dynamometer (Biodex 3, 20 Ramsay Rode, Shirley, New York, USA)^®^.	95% IC Pre-test, post-test: Balance and stabilityIG1: *p* < 0.001 IG2: *p* > 0.05 IG1 vs. IG2: *p* > 0.05 Isokinetic strength IG1: 60°: *p* < 0.001, in all tests. 180°: Quadriceps DL and NDL, *p* < 0.001 Hamstrings DL, *p* = 0.04; NDL. *p* = 0.1 300° Quadriceps DL and NDL, *p* < 0.001 Hamstrings DL, *p* = 0.002; NDL, *p* = 0.4 IG2: 60° Quadriceps DL and NDL *p* > 0.05 Hamstrings DL *p* = 0.015; NDL *p* = 0.026 180° Quadriceps DL *p* = 0.005; NDL, *p* = 0.003 Hamstrings DL and NDL, *p* < 0.001 300° Quadriceps DL *p* = 0.004; NDL, *p* = 0.002 Hamstrings DL *p* = 0.010; NDL *p* = 0.010
	IC: NS. EC: NS.		Balance, Stability, Concentric Isokinetic Strength for quadriceps/ hamstrings at 60,180 and 300°/s: dominant leg (DL) and non-dominant leg (NDL)	
Van der Horst et al. [19]. (2015). To investigate the preventive effect of the NHE on the incidence and severity of hamstring injuries in male amateur football players.	*n* = 579. (CG: 292; IG: 287) Male, health. 2013 season. Age = 24.5 ± 3.8 years.	CG (292) IG (287): NHE. 25 sessions/13 weeks.	Ad hoc registration form	95% CI Pre-test vs. Post-test No injuries: 38 injuries in 36/579 players (6.2%). CG: 0.8 (0.61–1.15) IG: 0.25 (0.19–0.35) CG vs. IG *p* = 0.005 Severity of the injury: No statistically significant
	IC: Dutch male amateur (high level) football players, 18–40 years. EC: joined a team after the start of the trial.		No injuries, severity	
Naclerio et al. [29]. (2015). To investigate the effects of two different 6-week lower body injury prevention programs on knee muscle torque–angle relationship was examined in football players.	*n* = 32 (CG:11, IG1:11, IG2:11). Male, health. Age = 22 ± 2.6 years. Weight = 75.9 ± 7.3 kg. Height = 178.9 ± 7.7 cm.	CG IG1: Eccentric knee flexors exercises, NHE included. IG2: Proprioceptive exercises. 18 sessions/6 weeks.	Load cell and I METRIC V. 8.32 software^®^ (Globus, Italy) for maximum voluntary isometric contraction test.	95% CI *Pre-test vs. Post-test* IG1 35° (t (29) = 2.227, *p* =0.034, d = 0.67) 45° (t (29) = 3.177, *p* = 0.004, d = 0.96) IG2 60° (t (29) = 3.836, *p* = 0.024, d = 1.16) 80° (t (29) = 4.027, *p* =0.018, d = 1.21) 90° (t (29) = 4.567, *p* =0.001, d = 1.38) Between-groups differences CG vs. IG2, *p* = 0.215 CG vs. IG1, *p* = 0.392 IG2 vs. IG1, *p* = 0.634 Not statistically significant
	IC: NS. EC: lower body resistance training programs in the 6 preceding months or a previous lower limb injury.		Maximum voluntary isometric force in knee flexion 35°, 45°, 60°, 80°, 90°, and 100° (hamstrings)	
Silvers-Granelli et al. [16]. (2015). To examine the efficacy of the FIFA 11+ program in men’s collegiate United States National Collegiate Athletic Association (NCAA) Division I and Division II football.	*n* = 1525 (CG: 850, IG:675). Male, health. Age = 18–25 years.	CG IG:11+ Running (8 min) with cutting, change of direction, decelerating, and proper landing; strength, plyometric, and balance exercises (10 min) focusing on core strength, eccentric control, and proprioception; and lastly, running (2 min) to conclude the warm-up 3/week, during a season.	HealtheAthlete injury surveillance system (web), where trainers included the data (registration form)	95% CI Incidence Rate CG, 665 injuries (mean ± SD, 19.56 ± 11.01) 15.04 injuries/1000 h IG, 285 injuries (mean ± SD, 10.56 ± 3.64) 8.09 injuries/1000 h IG vs. CG:0.54 [0.49–0.59]; *p* < 0.0001) Time loss CG (mean ± SD, 13.20 ± 26.6 days) IG (mean ± SD, 10.08 ± 14.68 days) IG vs. CG (*p* = 0.007).
	IC: student athletes participating in an NCAA Division I or II institution member EC: an injury prevention program in the past 4 competitive seasons.		Incidence Rate, time loss due to injury	
Sebelien et al. [30]. (2014). To examine if Nordic hamstring exercises (NHE) decreased injury rates, increased sprinting speed, and increased hamstring and quadriceps muscle strength among semiprofessional football players.	*n* = 142 (CG:70, IG:72). Male, health. Age = 18–39 years.	CG IG: individual stretches (3 sets/20 s), stretches with the partner (3 sets/45 s) and NHE (3 sets x12 repetitions). During 10 months.	Cybex 6000 (Lumex and Ronkonkoma, NY)^®^, isokinetic. 40-meters sprint test. Ad hoc registration form.	95% CI, Pre-test vs. post-test No of injuries during the program. IG vs. CG x^2^ (1) = 6.44, *p* = 0.010). Sprint speed in 10 meters of sprint. (t (13) = 3.43), *p* = 0.005, [−0.040, −0.009] No statistically significant Strength (eccentric and isometric isokinetic hamstring, and concentric isokinetic hamstring/quadriceps) Sprint speed in 30 and 40-m.
	IC: a hamstring injury currently or in the last 6 months, or other injuries preventing players to do initial strength and sprint testing protocols; hamstring injuries during the season preventing them to continue with the NH or their usual warm-up exercises, or to complete football practices or games for 2 weeks.		No of injuries, sprint speed, strength (eccentric and isometric isokinetic hamstring, and concentric isokinetic hamstring/quadriceps)	
Naclerio et al. [31]. (2013). To investigate the effects of a 4 weeks lower body injury prevention program on knee muscle torque–angle relationship in football players.	*n* = 20 (CG:10, IG:10) Male, health. Age = 23.8 ± 3.1 years. Weight = 756.8 ± 5.9 kg. Height = 176 ± 4.9 cm.	CG IG: Resistance program. Three exercises with eccentric force (NHE) and proprioception for 3 months.	Load cell and I METRIC V. 8.32 software^®^ (Globus, Italy) for maximum voluntary isometric contraction test.	95% CI Pre-test vs. Post-test Only 80° was significant (*p* = 0.001; d = 0.94)
	IC: NS. EC: any lower body resistance training during the 3 months prior to the study, or previous lower limb injury.		Maximum voluntary isometric force in knee flexion at 35°, 45°, 60°, 80°, 90°, and 100° (hamstrings).	
Daneshjoo et al. [15].(2013). To investigate the effects of eight-week 11+ and Harmoknee injury prevention training programs on the strength of the quadriceps and hamstrings in professional male football players.	*n* = 36 (CG = 12; IG1 = 12; IG2 = 12) Male, health. Age = 18.9 ± 1.4 years. Weight = 73.6 ± 6.3 kg. Height = 181.3 ± 5.5 cm.	CG: normal warm-up with stretching. IG1: 11+. 20–25 min sessions, 3/week IG2: Harmoknee 5 parts: warm-up, muscle activation, balance, strength, and core stability. 20–25 min sessions, 3/week, 8 weeks (24 sessions).	Biodex Isokinetic Dynamometer (Biodex 3, 20 Ramsay Rode, Shirley, New York, USA)^®^ for isokinetic test.	95% CI Pre-test vs. Post-test Concentric quadriceps force IG1 (*p* < 0.05) 300°·s^−1^ DL: increases 27% IG2 (*p* < 0.05): 60°·s^−1^ DL: increases 36.6%, 180°·s^−1^ DL: increases 36.2% 300°·s^−1^ DL: increases 28% 60°·s^−1^ NDL: increases 31.3%, 180°·s^−1^ NDL: increases 31.7% 300°·s^−1^ NDL: increases 20.05% Concentric hamstring force IG1/IG2 (*p* < 0.05): 60°·s^−1^ DL: increases 22%/32.5%, 180°·s^−1^ DL: increases 21.4%/31.3% 300°·s^−1^ DL: increases 22.1%/14.3% 60°·s^−1^ NDL: increases 22.3%/21.1%, 180°·s^−1^ NDL: increases 15.7%/19.3% Comparison of strength between groups IG1 vs. CG DL (*p* = 0.01); NDL (*p* = 0.02).
	IC: at least 5 years’ experience EC: history of major lower limb injury or disease		Hamstrings and quadriceps concentric isokinetic strength (bilateral) at 60°·s^−1^, 180°·s^−1^ and 300°·s^−1^	
Daneshjoo et al. [12]. (2012). To investigate the effect of FIFA 11+ and Harmoknee injury preventive warm-up programs on CSR, DCR and FSR in young male professional football players. These ratios are related to the risk of injury to the knee in football players.	*n* = 36 (CG = 12; IG1 = 12; IG2 = 12) Male, health. Age = 18–21 years. Weight = 62–83 kg. Height = 172–187 cm.	CG: normal warm-up with stretching. IG1: 11+. 20–25 min sessions, 3/week IG2: Harmoknee 5 parts: warm-up, muscle activation, balance, strength, and core stability. 20–25 min sessions, 3/week, 8 weeks (24 sessions).	Biodex Isokinetic Dynamometer (Biodex 3, 20 Ramsay Rode, Shirley, New York, USA)^®^ for isokinetic test.	95% CI Pre-test vs. Post-test IG1 CSR increase 8% at 60 u.s21 (t = 3.08, *p* = 0.01). DCR decrease 30% (*p* = 0.05) FSR increase 8% (t = 2.37, *p* = 0.03). IG2 No statistically significant CSR, FSR and CSR. Comparison between groups CSR IG1 vs. IG2: NDL at 60 u.s21 (F2.32 = 4.1, *p* = 0.02). DCR IG1 vs. CG: NDL leg (*p* = 0.04) IG2 vs. CG: NDL (*p* = 0.04) FSR Not statistically significant
	IC: NS. EC: history of major lower limb injury or disease.		Hamstrings and quadriceps concentric isokinetic strength (bilateral) at 60°·s^−1^, 180°·s^−1^ and 300°·s^−1^: measure CSR, DCR FSR.	
Petersen et al. [9]. (2011). To investigate the preventive effect of eccentric strengthening of the hamstring muscles using the Nordic hamstring exercise compared with an additional hamstring exercise on the rate of acute hamstring injuries in male football players.	*n* = 942 (CG:481, IG:461) Male, health. Age = 18–39 years. Follow-up of an entire season (2008–2009)	CG IG: NHE. 27 sessions for 10 weeks.	Injury registration form recommended by the Medical Assessment and Research Centre of the FIFA.	95% CI Pre-test vs. Post-test IG vs. CG Number of total injuries. [RR], 0.293; (0.150–0.572; *p* = 0.001) Recurring injuries. [RR], 0.137; (0.037–0.509; *p* = 0.003) New injuries. [RR], 0.410; (0.180–0.933; *p* = 0.034)
	IC: men’s football teams play in the top 5 Danish football divisions EC: NS.		No of total, recurring and new injuries.	

*n*: Total Sample, CG: Control Group, IC: Inclusion criteria, IG: Intervention Group, EC: Exclusion criteria, NS: Not specified, DL: Dominant leg, NDL: Non-dominant leg:, NHE: Nordic Hamstring exercise, NCAA: United States National Collegiate Athletic Association, 11+ (FIFA 11+): Federation International de Football Association, CI: Confidence Interval, CSR: Conventional strength ratio, CDR: Dynamic control ratio, FSR: Fast/slow speed ratio, Flex: Flexion, BEP: Bounding Exercise Program, NWP: New Warm-up Program [RR] adjusted ratio. FIFA: Federation International de Football Association.

**Table 2 jcm-10-02029-t002:** Completed PEDro quality appraisal.

Study	Criteria											Total Score
	1	2	3	4	5	6	7	8	9	10	11	
Van de Hoef et al. [27]	✔	✔	X	✔	X	X	X	✔	✔	✔	✔	6
Ghareeb et al. [28]	✔	X	X	✔	X	X	X	✔	✔	✔	✔	5
Van der Horst et al. [19]	✔	✔	✔	✔	X	X	X	✔	✔	✔	✔	7
Naclerio et al. [29]	✔	✔	X	✔	X	X	X	✔	✔	✔	✔	6
Silvers-Granelli et al. [16]	✔	✔	X	X	X	X	X	X	✔	✔	✔	4
Sebelien et al. [30]	✔	✔	✔	X	X	X	X	X	✔	✔	✔	4
Naclerio et al. [31]	✔	✔	X	✔	X	X	X	✔	✔	✔	✔	6
Daneshjoo et al. [15]	✔	✔	X	✔	X	X	✔	✔	✔	✔	✔	7
Daneshjoo et al. [12]	✔	✔	X	✔	X	X	✔	✔	✔	✔	✔	7
Petersen et al. [9]	X	✔	X	✔	X	X	X	✔	✔	✔	✔	6

Criteria: 1 Eligibility criteria were specified (not used for score); 2 Subjects were randomly allocated to groups; 3 Allocation was concealed; 4 Groups were similar at baseline regarding the most important prognostic indicators; 5 There was blinding of all subjects; 6 There was blinding of all therapists who administered the therapy; 7 There was blinding of all assessors who measured at least one key outcome; 8 Measures of at least one key outcome were obtained from more than 85% of the subjects initially allocated to groups; 9 All subjects for whom outcome measures were available received the treatment or control condition as allocated or, where this was not the case, data for at least one key outcome was analyzed by ‘intention-to-treat’; 10 The results of between-group statistical comparisons are reported for at least one key outcome; 11 The study provides both point measures and measures of variability for at least one key outcome.

**Table 3 jcm-10-02029-t003:** Significance and effectiveness of prevention protocols regarding different grouping variables.

	PREVENTION PROGRAMS								
	With NHE						Without NHE		
GROUPING VARIABLES	NHE	11+	Harmoknee	NHE + Stretches	NHE + Eccentric	NHE + Proprioceptives	Proprioceptives	NWP	BEP
Injuries incidence (hamstring): new or recurrent	✔ (9.19)	✔ (16)		✔ (30)					X (27)
Severity injuries (hamstring)	X (19)								X (27)
Time loss due to injury		✔ (16)							
Balance		X (28)						✔ (28)	
Stability		X (28)						✔ (28)	
Isokinetic strength (concentric)		✔ * q (28) ✔ h (28) ✔ q-h (12) ✔ * q-h (15)	X q-h (9) ✔ q (15) ✔ * h (15)	X q-h (30)				✔ q (28) ✔ * h (28)	
Isokinetic strength (eccentric)				X q-h (30)					
Isometric strength				X q-h (30)	✔ * h (29)	✔ * h (31)	✔ * h (29)		
Speed				✔ * (30)					
Total significant data ✔	1/2	5/7	1/2	2/5	1/1	1/1	1/1	3/4	0/2

Abbreviations: ✔ (Significant); ✔ * (Partially Significant. At least one variable was significant); X (Non-significant; 11+ (FIFA 11+ Federation International de Football Association); BEP (Bounding Exercise Program); h (hamstrings); NHE (Nordic Hamstring Exercise); NWP (New Warm-up Program); q (quadriceps).

## Data Availability

Data sharing not applicable. No new data were created or analyzed in this study. Data sharing is not applicable to this article.

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
