# Peer review of "Acute Hamstring Injury Prevention Programs in Eleven-a-Side Football Players Based on Physical Exercises: Systematic Review"

_jcm, 2021, doi:10.3390/jcm10092029_

Round 1
Reviewer 1 Report
I think that all prevention programs could be more or less effective for reducing acute hamstring injuries. I wonder if each prevention program had each specific features in itself? or had some overlaps among the programs? The manuscript did not touch why the program was effective? What's the differences between the effective programs and the ineffective ones?
Table 1: As a table, it is quite complicated to see and understand the whole picture of references. Please improve.
Reviewer 2 Report
The present systematic review aimed to analyze the type and the effectiveness of exercise programs to prevent acute hamstring injuries in soccer players. The study covers a hot topic that is worth being investigated and shared with the scientific community. The manuscript is well-written and composed, easy-to-read, and provides a detailed overview of the exercise programs used as prevention strategies. I would like to congratulate the Authors on their work.
Reviewer 3 Report
The purpose of the manuscript “Acute Hamstring Injury Prevention Programs in Eleven-A-Side Soccer Players Based on Physical Exercises: Systematic Review” was to analyze the exercise programs used for the prevention of acute hamstring injuries in soccer players 11, as well as their effective-ness. Unfortunately, this manuscript presents limitations and weakness. Grammar and sentence structure have been revised.
Abstract: revise. You have to highlight the results of studies and no number of studies founded. Conclusions are vague, please improve to give to the author a strong take-home message.
Introduction: revise. I think that could be useful to explain the risk factors in soccer
Discussion and Conclusions: revise. Please, improve the recommendations to practical application of this study.
Please, put on only 'soccer' or 'football' and no 'soccer 11' or 'football 11' in all text.
